TROPICAL DISEASES

# Rift Valley fever virus targets the maternal-foetal interface in ovine and human placentas

**Judith Oymans[1,2], Paul J. Wichgers Schreur[1], Lucien van Keulen[1], Jet Kant[1], Jeroen Kortekaas[1,2]***

1 Wageningen Bioveterinary Research, Houtribweg, Lelystad, The Netherlands, 2 Laboratory of Virology, Wageningen University & Research, Wageningen, The Netherlands

* Jeroen.kortekaas@wur.nl

**Data Availability Statement:** All relevant data are within the manuscript and its Supporting Information files.

## Abstract

### Background

Rift Valley fever virus (RVFV) is an arbovirus of the order *Bunyavirales* that causes severe disease in ruminants and humans. Outbreaks in sheep herds are characterised by newborn fatalities and abortion storms. The association of RVFV infections with abortions of ovines and other ruminants is well recognized, whereas the pathology resulting in abortion has remained undescribed. Accumulating evidence suggests that RVFV is abortogenic in humans as well, warranting more research on the interaction of RVFV with the ruminant and human placenta.

### Methodology/Principal findings

Pregnant ewes were inoculated with a highly virulent strain of RVFV and necropsied at different days post infection. Tissues were collected and analysed by PCR, virus isolation, and immunohistochemistry. The results show that RVFV replicates efficiently in maternal placental epithelial cells before the virus infects foetal trophoblasts. Moreover, the virus was shown to bypass the maternal epithelial cell layer by directly targeting foetal trophoblasts in the haemophagous zone, a region of the ovine placenta where maternal blood is in direct contact with foetal cells. Abortion was associated with widespread necrosis of placental tissues accompanied with severe haemorrhages. Experiments with human placental explants revealed that the same virus strain replicates efficiently in both cyto- and syncytiotrophoblasts.

### Conclusions/Significance

This study demonstrates that RVFV targets the foetal-maternal interface in both ovine and human placentas. The virus was shown to cross the ovine placental barrier via two distinct routes, ultimately resulting in placental and foetal demise followed by abortion. Our finding that RVFV replicates efficiently in human trophoblasts underscores the risk of RVFV infection for human pregnancy.

**Funding:** This work was commissioned and financed by the Dutch Ministry of Agriculture, Nature and Food Quality of the Netherlands, project code: WOT-01-001-003. The funders had no role in study design, data collection and analysis, decision to publish, or preparation of the manuscript.

**Competing interests:** The authors have declared that no competing interests exist.

## Author summary

Rift Valley fever virus (RVFV) is a mosquito-borne RNA virus that causes severe disease in ruminants, wildlife and humans in Africa and the Arabian Peninsula. Outbreaks are characterised by high mortality rates among newborn lambs and abortion storms in sheep herds. The severe outcome of RVFV infection during pregnancy in livestock is well documented, whereas the pathological changes that result in abortion have not yet been described. To investigate how RVFV crosses the placenta and how infection results in abortion, pregnant ewes were infected with RVFV and target cells in maternal and foetal tissues were identified at different time points after inoculation. We show that epithelial cells of the ovine placenta and foetal trophoblasts are primary target cells of RVFV and that placental demise is the primary cause of abortion. The same RVFV strain was shown to replicate efficiently in human placental explants, targeting both cyto- and syncytiotrophoblasts.

## Introduction

Rift Valley fever virus (RVFV) is a negative-strand RNA virus of the family *Phenuiviridae* (former family *Bunyaviridae*), genus *Phlebovirus*. RVFV is transmitted by mosquitoes, predominantly by species of the genera *Aedes* and *Culex* [1]. The virus is pathogenic to domesticated and wild ruminants, of which sheep are the most susceptible. Newborn lambs may succumb within hours after onset of symptoms and seldom survive the infection [2,3]. The most typical pathological feature in lambs is severe necrosis of the liver. Specifically, necropsy of fatal cases reveals a swollen, pale liver with focal to widespread necrosis of hepatocytes. Direct and indirect consequences of liver necrosis include icterus, oedema, and hydrops ascites. Other organs, such as the spleen, heart, kidney and intestines may reveal haemorrhages and congested veins. Fatal cases are generally associated with haemorrhages and signs of shock. Susceptibility decreases with age, although mortality rates in adult sheep may be as high as 60% [4]. The most characteristic feature of RVF outbreaks in sheep herds are abortion storms, in which all pregnant ewes in a herd may abort [5].

Humans can be infected via mosquito bite, although most human cases are attributed to contact with contaminated animal tissues, particularly during the slaughtering of diseased animals [6]. The disease in humans generally follows a transient, febrile course with severe headache and muscle pains. Infection of the eye may induce temporal or permanent loss of vision resulting from retinal damage [7]. In a minority of patients, the infection progresses to encephalitis or haemorrhagic fever, of which the latter is often fatal [4,8,9]. A cross-sectional study of Sudanese women correlated RVFV infection with an elevated miscarriage rate [10]. Notably, experiments with second-trimester human foetal tissue explants recently revealed that human syncytiotrophoblasts are susceptible and permissive to RVFV replication [11]. These reports warrant further studies to determine if RVFV infections pose a perhaps underestimated risk for human pregnancy.

The high abortion rates in sheep herds suggest that RVFV crosses the ovine placenta extremely efficiently. The placenta of sheep, and that of other RVFV susceptible ruminant species, is very different from the human placenta. The human placenta has the shape of a single disque whereas the ovine placenta consists of placentomes, varying in number among the different ruminant species [12]. Placentomes are discrete areas of extensive villous interactions between the maternal epithelial cells of the uterus, the caruncle, and the foetal trophoblasts of

the allantochorion, the cotelydon, enabling efficient exchange of gases and nutrients between mother and foetus. In sheep and goats, pools of maternal blood are located in the crypts of the foetal villi, directly bordering the foetal trophoblast layer. At these so-called haemophagous zones, maternal erythrocytes are phagocytosed by trophoblasts as a principle source of iron for the developing foetus. Notably, this part of the ovine placenta resembles the haemochorial placenta of humans, in that maternal blood is in direct contact with foetal trophoblasts. However, the blood pools at the haemophagous zones in sheep and goats are filled with stagnant blood and therefore do not contribute significantly to the maternal-foetal exchange of nutrients, whereas in the human haemochorial placenta maternal blood flows along the foetal trophoblasts enabling efficient nutrient exchange.

The detrimental consequences of RVFV infection during ovine pregnancy were already reported after the first outbreak of RVF in 1930 and abortion storms have since then become a hallmark of RVF epidemics [2,3,13–15]. Nevertheless, the pathogenesis of RVFV-induced abortion in pregnant ewes was not yet described in literature. In the present study, we experimentally infected pregnant ewes at one third- or at mid-gestation and describe the pathogenic events that result in abortion of the ovine foetus. We furthermore investigated the susceptibility of human placental explants for the same RVFV strain.

## Materials and methods

### Cells and viruses

Culture media and supplements were obtained from Gibco unless indicated otherwise. Baby Hamster Kidney (BHK-21) cells were maintained in Glasgow minimum essential medium (GMEM) supplemented with 4% tryptose phosphate broth, 1% minimum essential medium nonessential amino acids (MEM NEAA), 1% antibiotic/antimycotic (a/a) and 5% foetal bovine serum (FBS), at 37°C with 5% $CO_2$. Vero-E6 cells were maintained in minimum essential medium (MEM) supplemented with 1% a/a, 5% FBS, 1% glutamine and 1% MEM NEAA, at 37°C with 5% $CO_2$. BHK-21 and Vero-E6 cells were purchased from ATCC.

A recombinant version of RVFV strain 35/74, originally isolated from the liver of a sheep during a RVFV outbreak in 1974 in the Free State province of South Africa, was used in the present work as described [16]. The titre was determined on BHK-21 cells as 50% Tissue Culture Infective Dose ($TCID_{50}$) according to the Spearman-Kärber algorithm [16].

### Ethics statements

Animal experiments were conducted in accordance with the Dutch Law on Animal Experiments (Wet op de Dierproeven, ID number BWBR0003081) and the European regulations (EU directive 2010/63/EU) on the protection of animals used for scientific purposes. The procedures were approved by the animal ethics committee of Wageningen Bioveterinary Research (WBVR) and the Dutch Central Authority for Scientific Procedures on Animals (permit number AVD401002017894).

Human placentas were obtained after caesarean section of healthy women. This material is regarded as medical waste and therefore does not fall under the scientific medical research law of the Netherlands and does not need approval from an institutional review board. All donors have given written consent and consent forms are stored in accordance with the Dutch privacy law.

### Experimental design pregnant ewe trial

At a conventional Dutch sheep farm, 30 Texel-Swifter mix breed ewes that had delivered healthy lambs before, were treated with progesterone sponges to synchronise pregnancies.

After removal of the sponges, ewes were naturally mated. Ultrasounds were performed at 6–7 weeks after mating to confirm gestation. Pregnant animals to be enrolled in the studies were subsequently transported to WBVR and allowed to acclimatize for one week before the start of the experimental period under biosafety level 3 (BSL-3) conditions.

At 55 days of pregnancy (experiment 1) or 78 days of pregnancy (experiment 2), animals were inoculated via intravenous (IV) route with $10^5$ $TCID_{50}$ of RVFV in 1 ml medium, or with 1 ml medium (negative control animals). Following challenge, animals were closely monitored for clinical signs, body temperatures were recorded, and EDTA blood samples were collected (Fig 1A). The ewes of experiment 1 were euthanized and necropsied at 6 days post infection and the ewes of experiment 2 were euthanized on day 4 or at abortion (day 7). Ewes and their foetuses were euthanized by intravenous administration of 50 mg/kg sodium pentobarbital (Euthasol, ASTfarma) and subsequent exsanguination. Foetuses were exsanguinated by severing of the umbilical cord after which foetal blood was collected in EDTA tubes. From the ewes, samples were taken from the liver and spleen. From the foetuses, samples were taken from the brain, spleen, leg muscle tissue and liver. Samples were also taken from the amniotic fluid, umbilical cord and the placentomes (3 per foetus). Samples were placed on dry ice and subsequently stored at -80˚C. Samples for histology and immunohistochemistry were fixed in 10% phosphate-buffered formalin for at least 48 h followed by routine processing into paraffin blocks.

## Infection of human placental explants

Full term placentas were obtained by caesarean section from healthy donors at the Isala hospital in Zwolle, the Netherlands. The placentas were transported on ice and placed in large petridishes with complete medium (40% Dulbecco's Modified Eagle Medium [DMEM], 40% F12 nutrient mixture, 10% FCS supplemented with 1% a/a). Chorionic villi were separated from the placenta and cut in 4x4 mm pieces, after which the samples were washed 3x with PBS + a/a. Placental explants were incubated with $2,5x10^5$ $TCID_{50}$ of RVFV in 1 ml complete medium, or in 1 ml complete medium (negative controls) in a 24-wells plate. Medium was removed 16 hpi, after which explants were washed 3x with PBS + a/a. Samples were collected at 1, 2 or 4 days post inoculation (dpi). Samples for quantitative reverse-transcription PCR (RT-qPCR) were stored at -80˚C. For virus isolation, 200 μl supernatant samples from each timepoint were pooled. For IHC, explants were fixed in 10% phosphate-buffered formalin for 48 h followed by routine processing in paraffin blocks. Each sample was analysed in quadruplicate. The 1 dpi timepoint was taken at 16 hpi after washing the explants. At 3 dpi, 1 ml of fresh medium was added to the 4 dpi samples.

## Detection of viral RNA

RNA was extracted from ovine plasma and organ samples. Briefly, organ suspensions were prepared by homogenising 0,3–1 g of tissue in an IKA Ultra Turrax Tube DT-20 containing 7 ml $CO_2$-Independent Medium (CIM) supplemented with 1% a/a. The suspensions were transferred to 15 ml Falcon tubes and cell debris was removed by centrifugation for 15 min at 4952 $x g$. Fifty μl Proteinase K (5 μg/ml, Sigma) was added to 200 μl of the plasma samples or organ suspensions. Next, 200 μl AL buffer (Qiagen), supplemented with 2 μl polyadenylic acid A (5 mg/ml, Sigma) was added, the samples were thoroughly mixed and incubated at 56˚C for 15 min. Subsequently, 250 μl 99% ethanol was added and RNA was isolated using the Qiagen RNeasy kit according to the manufacturer's protocol.

RNA was extracted from human placental explants by first homogenising the explants in 1 ml TRIzol Reagent (Invitrogen) in Lysing Matrix D tubes (MP Biomedicals) using the TeSeE™

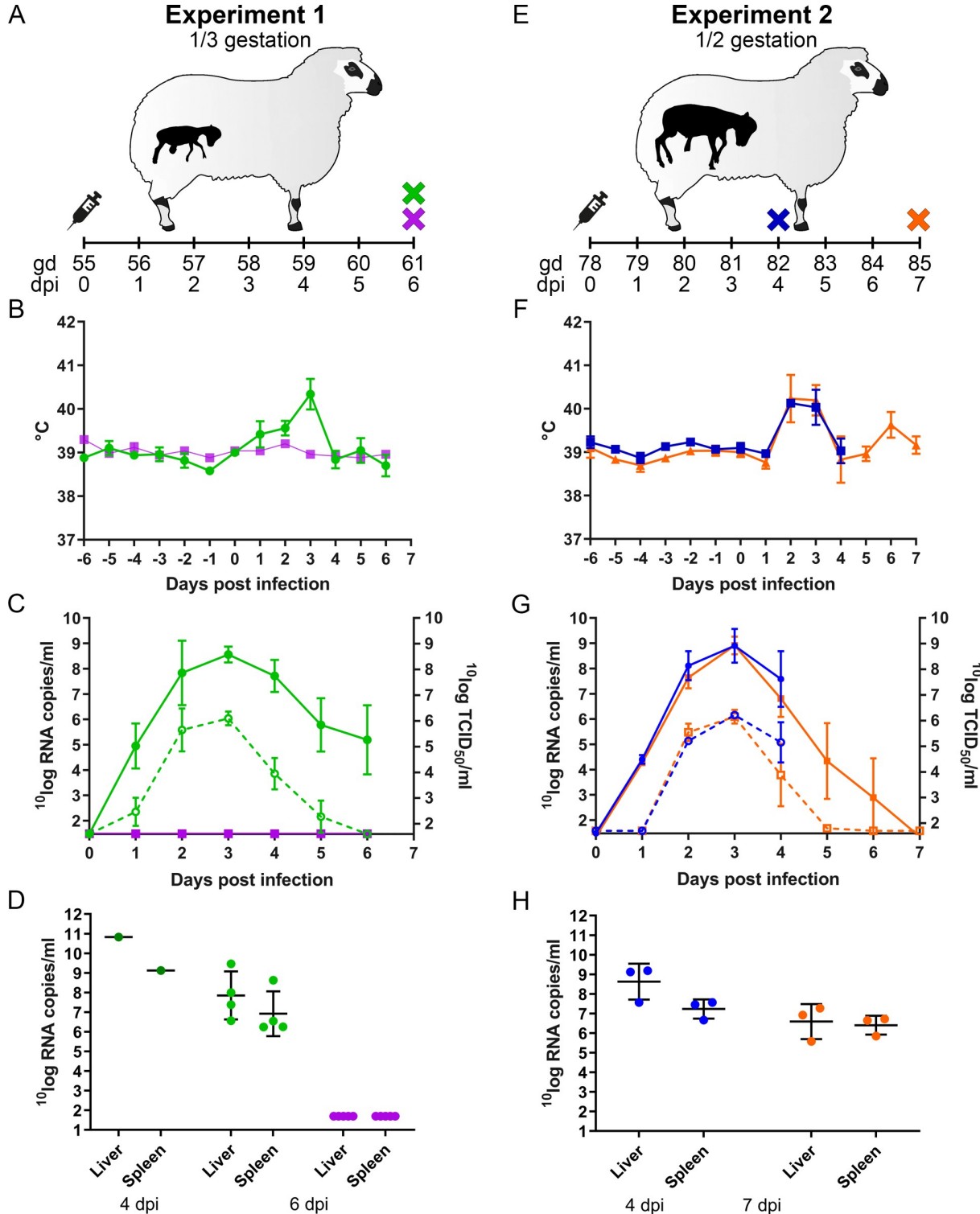

**Fig 1. Experimental design and primary outcome parameters ewes.** Ewes were inoculated intravenously with RVFV or mock inoculated at gestation day (gd) 55 (A) or day 78 (E) and euthanized at 4 (blue cross), 6 (green and purple crosses) or 7 (orange cross) days post inoculation (dpi). Purple numbers represent ewes that were mock-inoculated. Rectal temperatures (B, F), viremia by RT-qPCR (solid line, left y-axis), virus isolations (dotted line, right y-axis) (C, G) and the presence of RVFV in spleen and liver samples of the ewes (D, H) are depicted. Bars represent averages with SDs.

Precess 24 bead beater for 2x23 s at 6500 RPM. Supernatant (350 μl) was used for RNA isolation using the Direct-zol RNA miniprep kit (Zymo Research) according the manufacturer's protocol.

Five μl of the RNA was used in a RT-qPCR using the The LightCycler RNA Amplification Kit HybProbe (Roche, Almere, the Netherlands). Primers and probes were purchased from IDT. Forward primer: 5'-AAAGGAACAATGGACTCTGGTCA-3', reverse primer: 5'-CACTTCTTACTACCATGTCCTCCAAT-3'; Probe: 5'-6FAM-AAAGCTTTGATATCTCT CAGTGCCCCAA-TMR-3'. Cycling conditions were as follows: 45˚C for 30 min, 95˚C for 5 min, 45 cycles of 5 s at 95˚C and 35 s at 57˚C, followed by cooling down to 30˚C.

## Virus isolation

Virus isolations from ovine samples were performed by serial dilution of either plasma in complete CIM supplemented with 3,5 IU/ml heparin, or organ suspension in CIM, followed by incubation with BHK-21 cells. After 1.5 h incubation at RT, the inoculum was replaced by fresh medium and after 5 days of culturing the cells at 37˚C cytopathic effects were scored.

Virus isolations from the supernatant of the human placenta explants were performed by a 10x serial dilution series of the supernatant in complete medium on Vero cells. At 24 hpi, infection was visualized with an immunoperoxidase monolayer assay (IPMA). Briefly, cells were fixed and permeabilized 24 hpi with 4% paraformaldehyde (10 min) and ice-cold methanol (5 min). After permeabilization the plates were incubated with RVFV specific monoclonal antibody 4-D4, which recognizes the Gn protein [17]. A polyclonal rabbit-α-mouse immunoglobin/HRP antibody (Dako, Denmark) was used as a secondary antibody and 3-Amino-9-ethylcarbazole (AEC; Sigma-Aldrich) was used as a substrate. The titre was expressed as $TCID_{50}$/ml according to the Spearman-Kärber algorithm [18,19].

## Histology and immunohistochemistry

Paraffin embedded tissue was cut into 4 μm sections, collected on silane-coated glass slides and dried for at least 48 h in a 37˚C incubator. After deparaffinization and rehydration in graded alcohols, sections were stained routinely with haematoxylin and eosin (H&E) or immunostained for RVFV antigen. For immunostaining, endogenous peroxidase was blocked for 30 min in methanol/$H_2O_2$ followed by antigen retrieval through 15 min autoclaving at 121˚C in pH 6 citrate buffer (Antigen unmasking solution, Vector Laboratories ). As RVFV Gn-specific primary antibody, monoclonal antibody 4-D4 was used. Specificity of the immunostaining was confirmed with 2 other mAbs directed against different proteins of RVFV. Mouse envision peroxidase (Dako, Denmark) was used as secondary antibody and diaminobenzidine (DAB; Dako, Denmark) as the substrate, according to the manufacturer's instructions. Immunostaining for cytokeratin was performed using a rabbit mAb to cytokeratin 19 (Abcam 52625, USA) followed by α-rabbit-ImmPRESS-AP and Vector Red as substrate (Vector Laboratories, USA). Hematoxylin was used to counterstain the slides.

## Results

### RVFV infection of pregnant ewes results in abortion within one week

To study the pathogenesis of RVF in pregnant ewes, two experiments were performed. Ewes were infected at either one third of gestation (experiment 1) or at mid gestation (experiment 2). In both experiments, ewes were inoculated via intravenous route with a dose of $10^5$ $TCID_{50}$ of RVFV, an exposure route and dose that was used in previous studies with sheep [20–22]. In the first experiment, a group of ten ewes was distributed randomly over two groups of five

animals of which the first group was inoculated with virus and the second group with medium only (mock) (Fig 1A). As expected, and similar as observed in lambs, all RVFV-inoculated ewes developed fever (Fig 1B), which correlated with viremia as determined by RT-qPCR and virus isolation on plasma samples (Fig 1C). At 4 days post infection (dpi), one of the RVFV inoculated ewes (animal number 1764) acutely succumbed to the infection. Necropsy revealed a swollen, discoloured liver and signs of shock. Analysis of liver and spleen samples by RT-qPCR revealed high levels of viral RNA in both organs (Fig 1D). Extensive haemorrhages were found in the placentas of the two foetuses carried by this ewe and in the uterine wall. The remaining ewes were euthanized and necropsied as scheduled at 6 dpi. All ewes showed a multifocal necrotizing hepatitis which was most severe in the animal that died at 4 dpi. Although no abortions had occurred, all placentas revealed extensive haemorrhages in the placentomes and all foetuses had already died (S1 Fig, S3 Fig and Table 1).

To study the pathology at an earlier and later stage than 6 dpi, another group of 6 ewes from the same herd was inoculated with RVFV (experiment 2). Of this group, three ewes were euthanized and necropsied at 4 dpi and the remaining three ewes when abortion was imminent (Fig 1E). Of note, these ewes were at the moment of inoculation at mid gestation. As expected, all ewes of this experiment developed similar rises in rectal temperatures and viremia levels as those of experiment 1 (Fig 1F and 1G). Necropsy at 4 dpi revealed multifocal necrotizing hepatitis in all ewes but no macroscopic abnormalities in placentas and unaffected, live foetuses. In the morning of day 7 post infection, one ewe had aborted 2 foetuses and a second one was is the process of aborting (2 foetuses already expelled, 1 foetus still in the uterus). Necropsy of the third ewe (#1846) revealed three foetuses that were still inside the uterus but with placentomes showing extensive haemorrhages. One of these foetuses was alive, whereas the remaining two foetuses were found dead (Table 1). Analysis of liver and spleen samples revealed very high viral RNA levels in the organs of ewes necropsied at 4 dpi and lower levels in ewes necropsied at 6 dpi or at abortion (Fig 1D and 1H). Viral RNA and viable virus were detected in all placentomes from both experiments (Fig 2). It was striking to observe that viral RNA levels increased in placentomes between days 4 and 7 (Fig 2), whereas viremia and viral RNA levels in spleens and livers declined in the same period (Fig 1D and 1H).

## RVFV infects maternal epithelial cells and foetal trophoblasts

The ovine placenta consists of multiple placentomes, in which foetal blood is separated from maternal blood by several cell layers (Fig 3A and 3B). In accordance with literature, we observed an increase in mass of the foetal and maternal villi with concomitant decrease in mesenchyme and expansion of haemophagous zones with progressing pregnancy (Fig 3C and 3D) [23].

Necropsy of the ewe of experiment 1 that died at 4 dpi (#1764) revealed macroscopic abnormalities: extensive haemorrhages in the uterine wall, at the base of the placentomes and within the placentomes. Necropsy of ewes that were euthanized at 4 dpi in experiment 2 revealed placentomes that did not show any macroscopic abnormalities.

Immunohistochemistry showed the presence of RVFV antigen as small foci spread throughout the placentomes collected at 4 dpi in experiments 1 and 2 (Fig 4A). These foci consisted of a cluster of strongly stained epithelial/syncytial cells of the maternal villus with occasionally a single positive trophoblast in the epithelial lining of the opposite foetal villus (Fig 4B). Clusters of RVFV-positive cells were also found in the trophoblast epithelium lining the haemophagous zone (Fig 4C).

At 6 dpi, extensive haemorrhages were observed in the placentomes of the RVFV infected ewes (S1 Fig). In addition, in some placentomes maternal and foetal villi were starting to

**Table 1. Analysis of foetuses collected from RVFV-infected ewes.** Pathological findings in foetuses of necropsied ewes experimentally infected with RVFV at one third (1/3 gestation) or at mid-gestation (1/2 gestation). Ewes were necropsied at 4, 6 or 7 dpi. Viral RNA and proteins were detected by RT-qPCR and IHC, respectively. NT; not tested as foetuses were too autolytic.

| One third gestation | DPI | Foetuses | Alive or dead at time of dissection | Placentome | | non-cotyledonary allantochorion | | Liver | | Brain | | Amniotic fluid | Umbilical cord | | Umbilical blood |
|---|---|---|---|---|---|---|---|---|---|---|---|---|---|---|---|
| | | | | PCR | IHC | PCR | IHC | PCR | IHC | PCR | IHC | PCR | PCR | IHC | PCR |
| | 4 | 1764[1]-F1 | Dead | + | + | + | + | + | - | - | - | + | + | - | NT |
| | | 1764[1]-F2 | Dead | + | + | + | + | + | - | + | - | + | + | - | NT |
| | 6 | 1760-F1 | Dead | + | + | + | + | + | + | + | +[2] | + | + | +[2] | NT |
| | | 1761-F1 | Dead | + | + | + | + | + | - | - | - | + | + | - | NT |
| | | 1761-F2 | Dead | + | + | + | - | + | - | - | - | + | + | - | NT |
| | | 1762-F1 | Dead | + | + | + | - | + | - | - | - | + | + | - | NT |
| | | 1762-F2 | Dead | + | + | + | + | + | - | + | NT | + | + | - | NT |
| | | 1762-F3 | Dead | + | + | + | + | + | + | + | NT | + | + | +[2] | NT |
| | | 1763-F1 | Dead | + | + | + | + | + | + | + | +[2] | + | + | +[2] | NT |
| | | 1763-F2 | Dead | + | + | + | + | + | + | + | +[2] | + | + | +[2] | NT |
| Mid-gestation | 4 | 1841-F1 | Alive | + | + | + | - | - | - | - | - | - | + | - | + |
| | | 1841-F2 | Alive | + | + | + | - | - | - | - | - | - | + | - | + |
| | | 1842-F1 | Alive | + | + | + | - | + | - | - | - | - | + | - | + |
| | | 1842–F2 | Alive | + | + | + | - | + | - | - | - | + | - | - | + |
| | | 1843-F1 | Alive | + | + | + | - | + | - | - | - | + | + | - | + |
| | 7 | 1844-F1 | Dead | + | + | + | + | NT | NT | NT | NT | NT | NT | NT | NT |
| | | 1844-F2 | Dead | + | + | + | + | NT | NT | NT | NT | NT | NT | NT | NT |
| | | 1844-F3 | Dead | + | + | + | + | + | + | NT | +[2] | NT | + | +[2] | NT |
| | | 1845-F1 | Dead | + | + | + | + | NT | NT | NT | NT | NT | NT | NT | NT |
| | | 1845-F2 | Dead | + | + | + | + | NT | NT | NT | NT | NT | NT | NT | NT |
| | | 1846-F1 | Alive | + | + | + | + | + | + | + | +[2] | + | + | +[2] | + |
| | | 1846-F2 | Dead | + | + | + | + | + | + | + | +[2] | + | + | +[2] | NT |
| | | 1846-F3 | Dead | + | + | + | + | + | + | + | +[2] | + | + | +[2] | NT |

[1] Ewe died at 4 dpi

[2] Blood vessels (blood and endothelium).

separate. Both maternal epithelial cells (viable and necrotic) and foetal trophoblasts were strongly positive for RVFV antigen throughout the entire placentome (Fig 4F). Compared to the placentomes at 4 dpi, the area of RVFV positive maternal epithelial cells and foetal

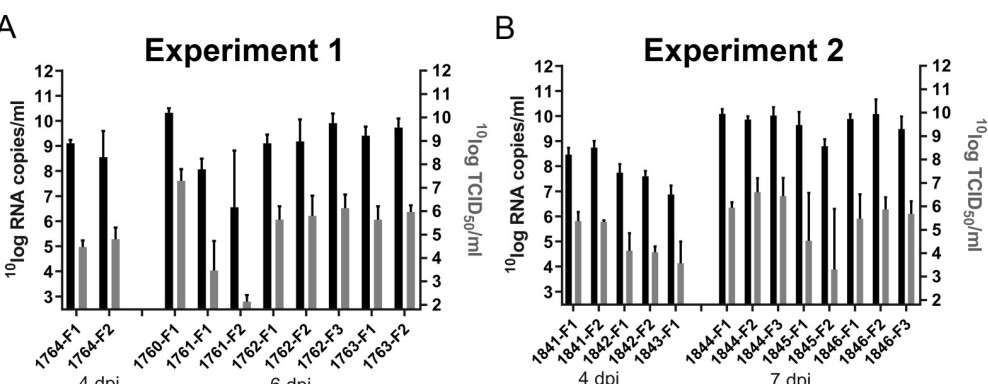

**Fig 2. Detection of viral RNA and infectious virus in placentomes.** Viral RNA copies in placentomes as determined by RT-qPCR (black columns; left y-axis) and virus titres in placentomes as determined by virus isolation (grey columns; right y-axis). Results of experiment 1 (A) and 2 (B) represent means and SDs of 3 placentomes per foetus.

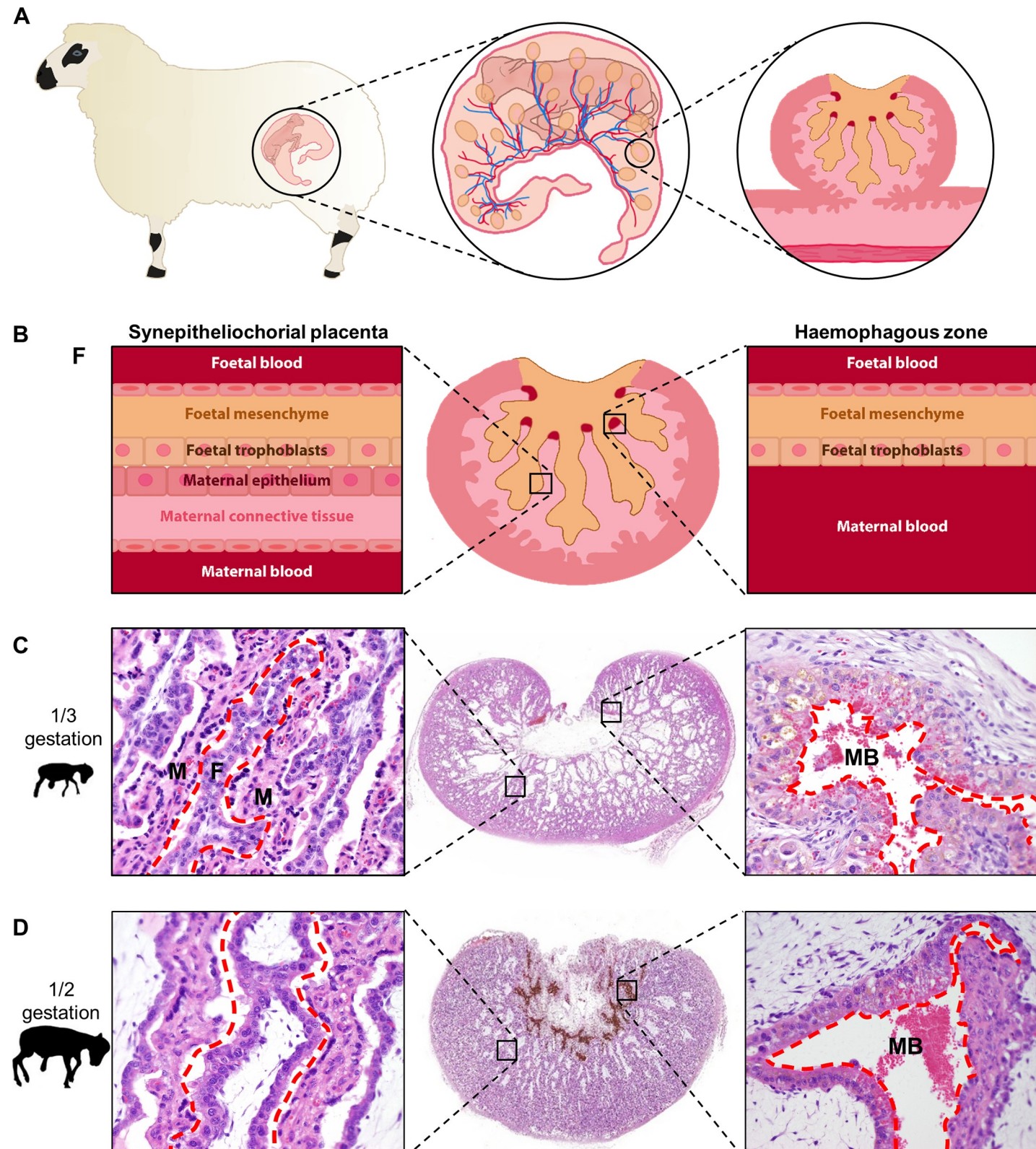

**Fig 3. The ovine placenta at different stages of gestation.** (A) Schematic presentation of an ovine foetus, the cotyledons and their blood supply. Only the foetal parts of the placenta (cotyledons) are displayed, the uterus wall and maternal part of the placenta (caruncles) are not depicted. At the right, a cross section of a placentome is

depicted, showing the maternal tissues in shades of pink, and the foetal villi in orange. Haemophagous zones at the base of the foetal villi are depicted in red. (B) A schematic overview of the cotyledon (center) with the different cell layers of the synepitheliochorial placenta at the left and the haemophagous zone at the right. In the synepitheliochorial placenta, the foetal blood is separated from maternal blood by several maternal and foetal cell layers. In the haemophagous zone maternal blood is in direct contact with the foetal trophoblasts. (C, D) HE staining of placentomes, the haemophagous zones and synepitheliochorial placenta at 1/3 gestation (C) and 1/2 gestation (D). Red interrupted lines indicate the boundaries between maternal and foetal tissues. F = foetal villus, M = maternal villus, MB = maternal blood. Notice the increase in the foetal/maternal villous interface in the synepitheliochorial placenta and the increase in size and erythrophagous activity of the trophoblasts of the haemophagous zone between 1/3 and 1/2 of the gestation period.

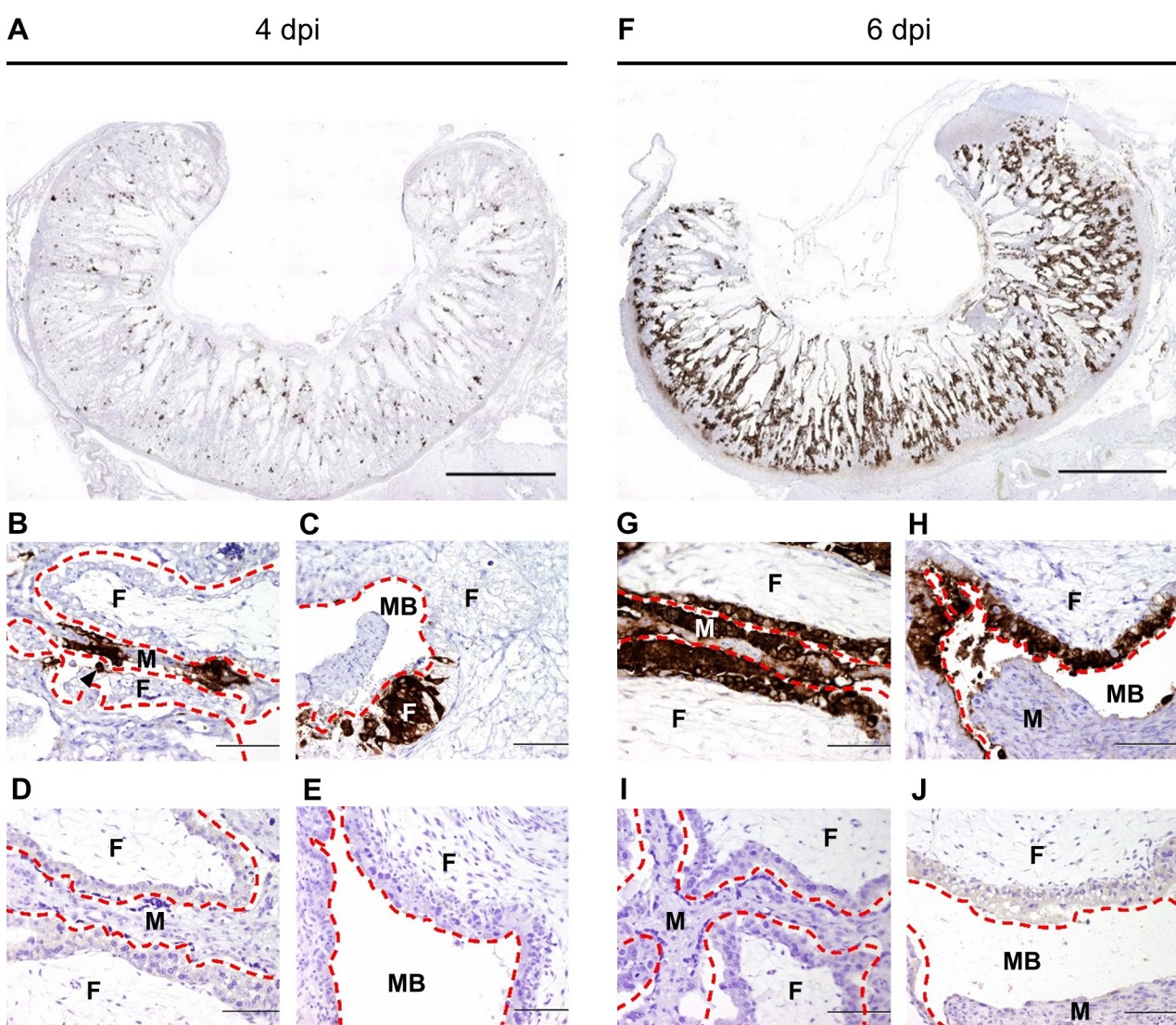

**Fig 4. RVFV antigen in placentomes.** Immunohistochemical (IHC) detection of RVFV antigen in cross sections of the placentomes at 4 dpi (experiment 2) and 6 dpi (experiment 1). At 4 dpi (A) only small foci of antigen positive cells are visible throughout the placentome while at 6 dpi (F) almost the entire placentome stains positive for RVFV antigen. Higher magnification of the synepitheliochorial placenta (SP) at 4 dpi (B) shows strong labelling of the maternal epithelial cells with only an individual positively stained foetal trophoblast (black arrowhead). At 6 dpi both maternal and foetal cell layers are strongly stained (G). In the haemophagous zone (HZ) at 4 dpi (C) only small clusters of foetal trophoblasts stain positive for RVFV antigen while at 6 dpi (H) the entire foetal trophoblast lining of the haemophagous zone stains positive. Panels D, E, I and J represent cross sections of uninfected placentomes corresponding to B, C, G and H, respectively, showing absence of background IHC staining. Red interrupted lines indicate the boundaries between maternal and foetal tissues. F; foetal villus, M; maternal villus, MB; maternal blood. Bar = 5000 μm (A, F) or 100 μm (B-J).

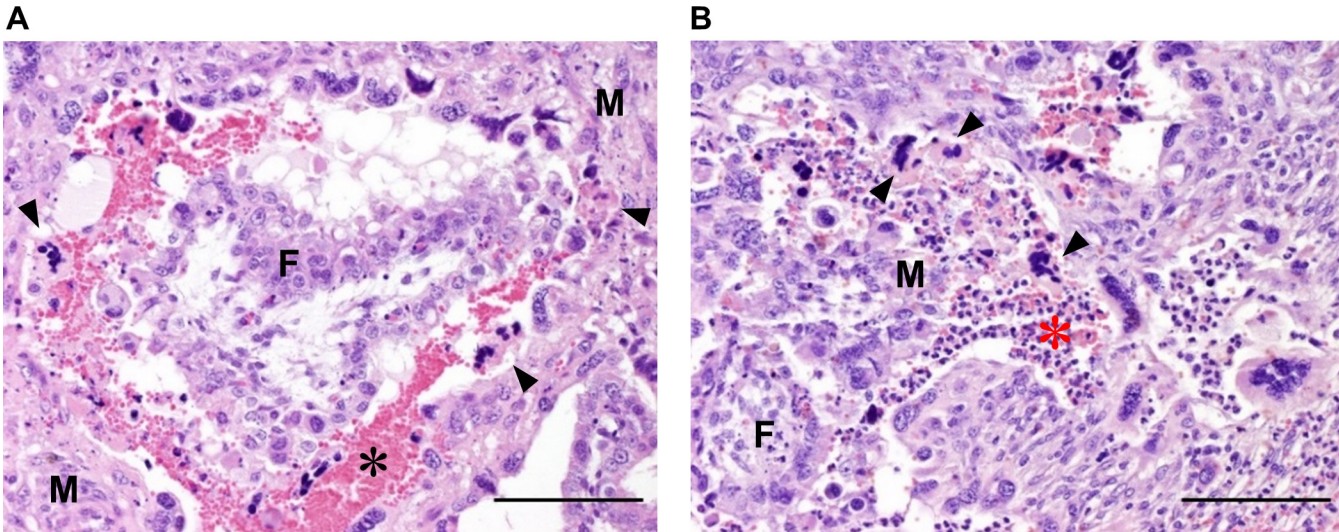

**Fig 5. Histopathology of the placentomes at imminent abortion.** HE staining of placentomes at 6 dpi. (A) Haemorrhages (black asterisk) and necrosis of maternal epithelial cells (arrowheads). Notice the relatively intact foetal epithelium. (B) Influx of neutrophils (red asterisk) and necrosis of maternal epithelium (arrowheads). F = foetal villus, M = maternal villus, Bar = 100 μm.

trophoblasts was greatly increased covering almost the entire maternal and foetal epithelial lining of the placentome including the haemophagous zone (Fig 4G and 4H). H&E staining revealed haemorrhages in the maternal villi with extensive necrosis of the maternal epithelium but only limited areas of necrosis of foetal trophoblasts (Fig 5A). In addition, a heavy influx of neutrophils was present mainly in the stratum compactum of the lamina propria and at the base of the maternal villi (Fig 5B).

At imminent abortion at 7 dpi, placentomes showed extensive haemorrhages with varying degrees of separation of maternal and foetal parts. Histology and immunohistochemistry revealed similar results as those obtained from analysis of the 6 dpi group with haemorrhages and a neutrophilic inflammatory response in maternal villi, necrosis of maternal epithelium and strong positive staining for RVFV in maternal epithelial cells and foetal trophoblasts. In placentomes where foetal villi had already separated from the maternal caruncle, large areas of denuded maternal villi were seen. In addition in some maternal villi, blood vessels were found that stained positively for RVFV in the endothelium and/or smooth muscle cells in the blood vessel wall (S2 Fig). Notably, endothelial cells did not reveal signs of apoptosis or necrosis.

## Detection of RVFV in foetal tissues

Ewe 1764 that acutely succumbed to the infection in experiment 1 carried two foetuses. RVFV RT-qPCR revealed viral RNA in the livers of both foetuses (Table 1 and Fig 6A). The foetuses of the ewes necropsied at 4 dpi in experiment 2 were all alive and appeared to be normal at the moment of necropsy (S3 Fig). However, RT-qPCR showed the presence of viral RNA in the blood of all foetuses and in the livers of 3 out of 5 foetuses (Fig 6B). These results show that RVFV is able to reach the foetus within 4 days.

All 8 foetuses of the ewes necropsied at 6 dpi were found dead within the uterus (Table 1). The livers of these foetuses were also positive for viral RNA and in 4 out of 8 foetuses viral antigen was detected by immunohistochemistry. In these 4 foetuses, the liver showed massive necrosis with only a few viable hepatocytes left (Fig 6C). Brain samples collected from 5 of the 8 foetuses were also found to contain viral RNA. Immunohistochemical staining of brain

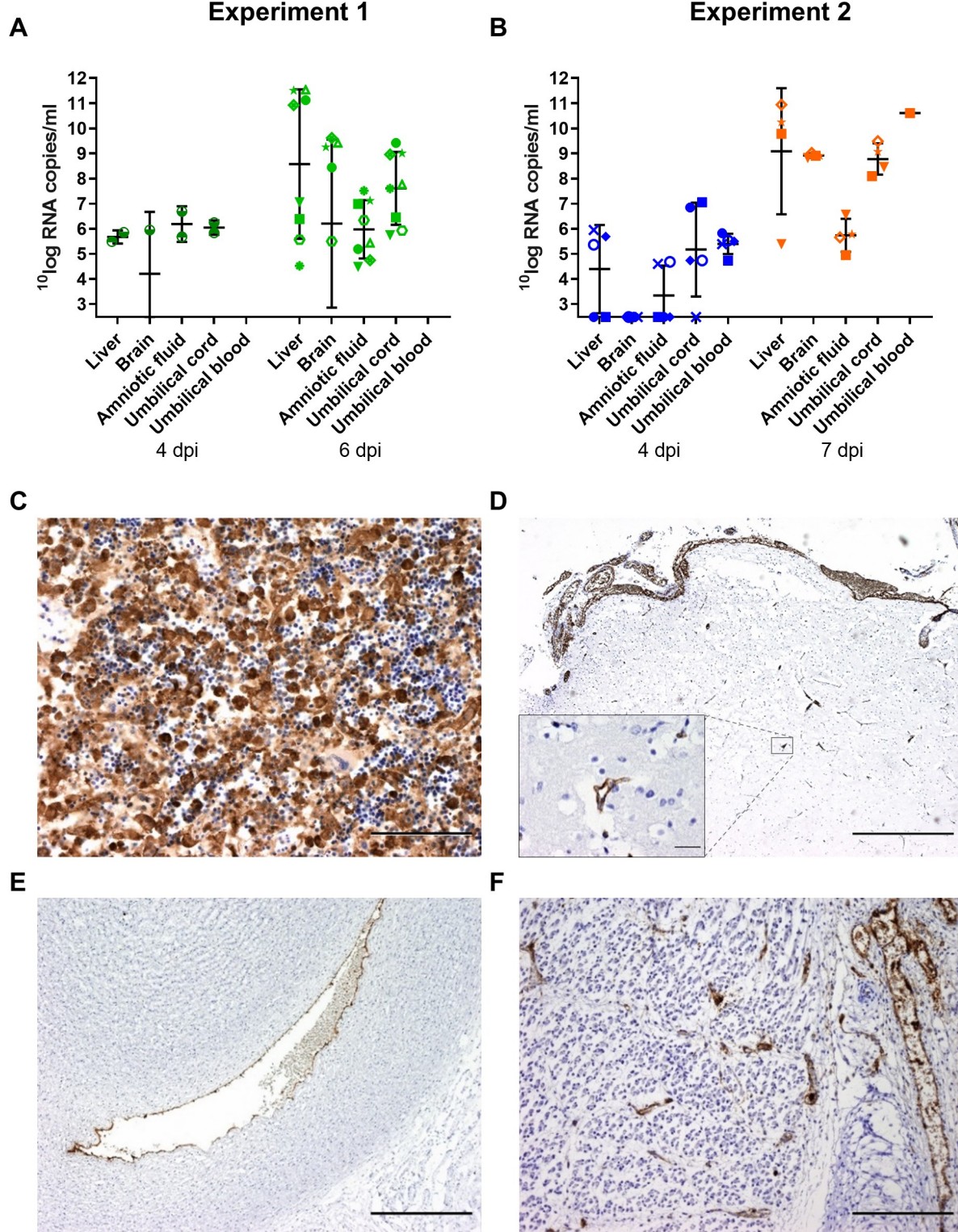

**Fig 6. Detection of viral RNA and viral antigen in foetal organs.** Detection of RVFV RNA by RT-qPCR in of organ suspensions of foetal organs collected in experiment 1 (A) and experiment 2 (B). Bars represent averages with SDs. Staining of RVFV antigen in samples collected from liver (C), brain (D) umbilical cord (E) and leg muscle (F). Notice the strong staining of endothelial cells in the blood vessels within the various organs. Bar = 500 μm (D, E, F), 100 μm (C) or 20 μm (inset D).

tissues from these foetuses revealed viral antigen in the blood and the endothelium of the blood vessels throughout the brain and arachnoidea (Fig 6D). No viral antigen was detected in neurons or glial cells. RVFV antigen was also detected in endothelial cells of blood vessels of the umbilical cord (Fig 6E) and muscle tissues (Fig 6F).

Most of the aborted foetuses in experiment 2 were severely autolytic and unsuited for further analysis (S3 Fig). The foetuses that were suitable for further analysis presented with severely necrotic livers and contained high levels of viral RNA in liver and brain samples. Hepatocytes and endothelial cells were strongly positive for RVFV antigen.

## RVFV strain 35/74 infects cytotrophoblasts and syncytiotrophoblasts in human placental explants

To study if RVFV strain 35/74, which was originally isolated from sheep, replicates in human placentas, explants of human term placentas were inoculated with the virus. The results show that the virus replicates efficiently in cytotrophoblasts and syncytiotrophoblasts (Fig 7), as revealed by RT-qPCR, virus isolation and IHC. In some areas of the placenta, viral antigen was detected in cytotrophoblasts but appeared to be absent in syncytiotrophoblasts, suggesting that the former are more permissive to RVFV replication (Fig 7C). The differences and similarities between the ovine and human placenta are depicted in S4 Fig.

## Discussion

The most characteristic feature of RVFV epizootics are abortion storms in sheep herds. During these events, all pregnant ewes in an affected herd may abort their foetuses. Although the detrimental effects of RVFV infection on ruminant pregnancy are well-recognized, the route that the virus uses to cross the placenta and the pathogenic events that result in abortion have remained undescribed. Here, we report that maternal and foetal epithelial cells in the ovine placenta are highly susceptible and permissive for RVFV and that abortion results from severe pathology of the placenta. In some ewes, the rapid progression of placental demise caused foetal mortality before foetuses could be infected.

During gestation, the progesterone hormone keeps the uterus and myometrium in a quiescent state to allow successful foetal development until parturition [24]. During the first trimester of gestation in sheep, progesterone is produced by the corpus luteum which resides in the ovaries. During the second and third trimester, the developing placenta becomes an additional source of progesterone. Placental progesterone is produced by binucleate trophoblasts that migrate to the maternal epithelium to fuse with uterine epithelial cells to form so-called syncytial cells. At about 90 days of pregnancy, half of the progesterone in the pregnant ewe is produced by the corpus luteum and half by the placenta. Our study has revealed that both trophoblasts and (syncytial) epithelial cells are major target cells of RVFV. Widespread necrosis of these cells likely results in a drop of systemic progesterone levels. In addition, virus-induced necrosis of placental cells leads to the production of pro-inflammatory chemokines and cytokines like TNF-α, IFN-γ and IL-8, which stimulate prostaglandin (E2/F2α) excretion by the placental epithelium. Prostaglandins bring the corpus luteum in regression, resulting in a further drop in progesterone levels. In the absence of the inhibitory effect of progesterone, prostaglandins induce uterine contractions and cervical effacement resulting in abortion. Inflammatory cytokines, most notably TNF-α, are also known to reduce vascular integrity. In the face of lowered coagulation factors in maternal blood resulting from liver necrosis, this could explain the observed placental haemorrhages.

Our study has revealed that RVFV uses at least two routes to cross the ovine placenta. The first route involves the haemophagous zones of the placenta, where foetal trophoblasts are in direct contact with maternal blood. Trophoblast cells in the haemophagous zone are

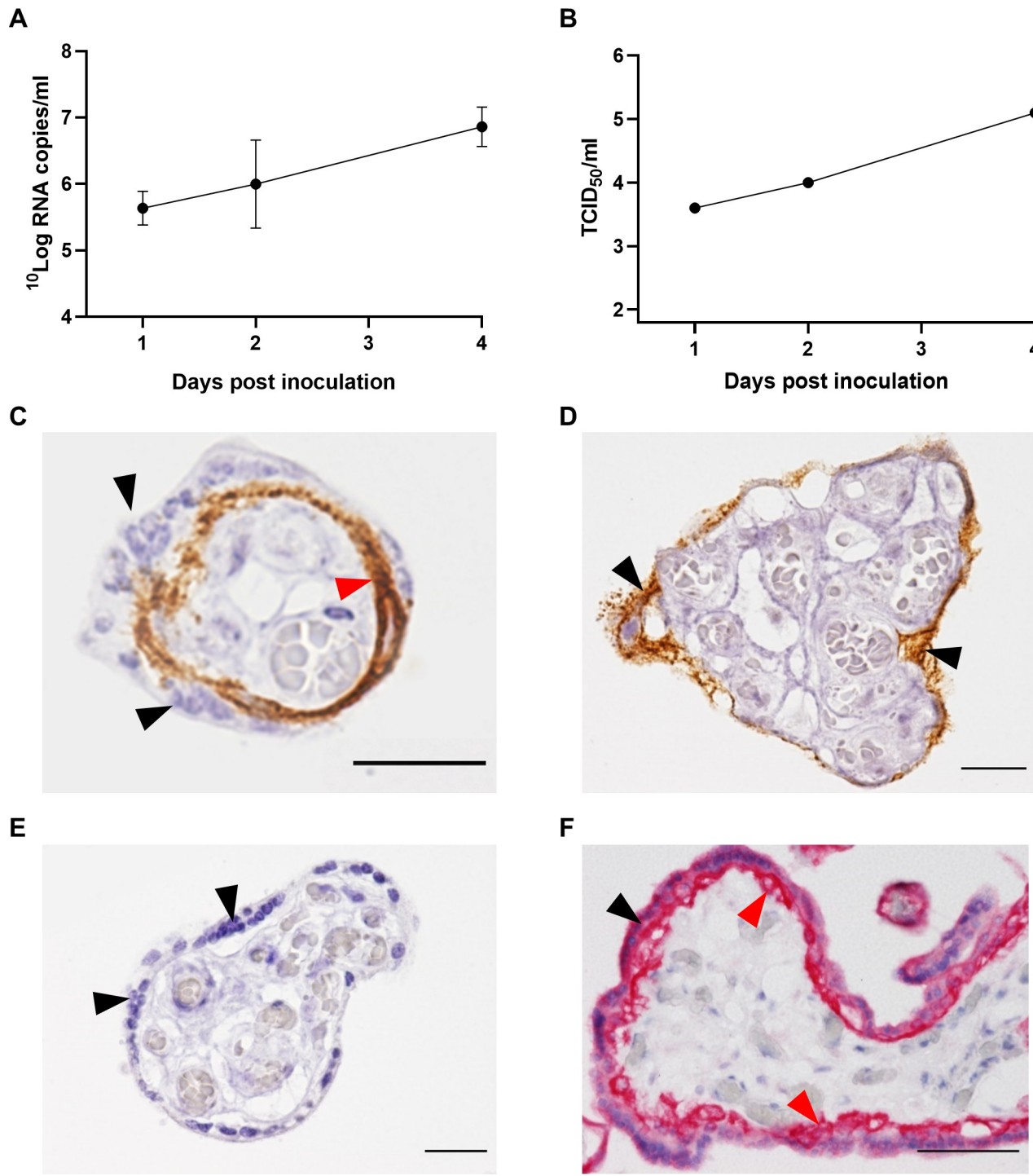

**Fig 7. RVFV in human placental explants.** Detection of viral RNA (A) and infectious virus (B) in human full term placental explants at different timepoints post infection. Viral RNA was detected by RT-qPCR and infectious virus by virus isolation. (C, D, E) Immunohistochemical staining of RVFV with mAb 4-D4, counterstained with haematoxylin. (C) Single villus with syncytiotrophoblasts (black arrowheads) staining negative for RVFV antigen with cytotrophoblasts (brown staining, red arrowhead) staining positive. (D) Single villus in which no cytotrophoblasts are present, with positive staining of syncytiotrophoblasts (black arrowheads). (E) Single villus showing absence of background IHC staining in non-infected control placental explant. Syncytiotrophoblasts are indicated with black arrowheads. (F) Immunohistochemical staining of epithelial cells in a non-infected placental villus with a mAb to cytokeratin showing both the syncytiotrophoblast layer (black arrowheads) and the cytotrophoblast layer (red arrowheads). Bar = 20 μm (C, D, E) or 50 μm (F).

specialised in phagocytosis of erythrocytes from the blood to provide the foetus with iron. The presence of RVFV in maternal pools of blood in the haemophagous zone may therefore result in direct infection of trophoblasts or via co-uptake of the virus during erythrophagy. The second route initiates with infection of maternal epithelial cells of the placenta. Virus released from these cells exposes foetal trophoblasts. Progeny virus produced by infected trophoblasts is subsequently released into the foetal mesenchyme exposing endothelial cells of foetal blood-vessels, which were also identified as target cells of RVFV.

The finding that foetal endothelial cells and endothelial cells in the maternal caruncle are target cells of RVFV is notable, as most of our previous studies with juvenile and adult sheep did not reveal endothelial cell infection. However, in one of our studies, RVFV-positive endothelial cells were detected in lymphoid organs of a lamb that peracutely succumbed after developing exceptionally high viremia [25]. In another study, endothelial cells of the skin became infected after feeding of mosquitoes on a viremic lamb [26]. These findings, together with the haemorrhagic manifestations that were seen in RVFV infected placentomes in this study, calls for further research into the interaction of RVFV with endothelial cells.

In a recently published study by McMillen and co-workers, RVFV infection of human placental explants was shown for the first time, using a strain that was originally isolated from humans [11]. Our experiments with human placental explants corroborate this research and highlight the potential risks of RVFV infection during human pregnancy. In the same study, the pathogenesis of RVFV infection in pregnant Sprague-Dawley rats was described. This work demonstrated that vertical transmission in these rats occurs through direct placental infection and that viral loads in the placenta were higher than in the liver and other maternal organs, similar as observed in the present study. In our study, the ovine placenta was not only found to contain the highest viral loads but was also the only organ still containing high levels of infectious virus at the moment of necropsy. These findings suggest that RVFV efficiently counteracts innate immune responses in placental cells.

RVFV counteracts host innate immunity through several functions of the nonstructural NSs protein [27–32]. One of the major functions of NSs is the downregulation of type I interferon (IFN) responses, which occurs through downregulation of general host gene transcription and the direct inhibition of IFN-β mRNA production [27,28,32]. Whereas in most cell types, type I IFNs play a major role in innate immunity, in cells of epithelial origin, such as the cells of the placenta, innate immunity is regulated by type III IFNs, referred to as IFN-λ [33]. Importantly, both type I and type III IFNs trigger the JAK/STAT pathway, which was shown to be targeted by NSs [34]. We therefore hypothesize that RVFV NSs facilitates replication in placental tissues by downregulating JAK/STAT signalling. On the other hand, previous studies have demonstrated that RVFV can also cross the ovine placental barrier without NSs [35]. These infections did not result in abortions but instead in stillbirths and congenital malformations, including arthrogryposis and hydranencephaly, resembling pathology in ovine foetuses infected by members of the genus *Orthobunyavirus*, such as Schmallenberg virus.

In conclusion, the present work has revealed how RVFV crosses the ovine placental barrier and has provided novel insights into the pathology that results in abortion in the most susceptible target species. The sheep isolate that was used was shown to replicate efficiently in human placental explants as well, calling for further research on the risk of RVFV infection during human pregnancy.

## Supporting information

**S1 Fig. RVFV infection results in extensive placental haemorrhages.** Placenta from a healthy ewe (A) and from an ewe inoculated with RVFV, necropsied six days after inoculation (B).

Placentas were collected during experiment 1 at one third of gestation.
(TIF)

**S2 Fig. Replication of RVFV in caruncles is associated with bleeding and infection of maternal endothelial cells.** (A) Micrograph of HE-stained caruncle tissue of ewe 1845 euthanized at 7 dpi (experiment 2). Notice the extensive haemorrhages at the base of the caruncle (arrowhead and inset). (B) Immunohistochemical staining of RVFV antigen. Positive staining is only seen in those areas where necrotic maternal epithelium is still present. Some maternal blood vessels are also stained (arrowhead and inset) and show the presence of RVFV antigen in the endothelial cells and the smooth muscle cells of the tunica media (inset). Bar = 1000 μm (A, B), 100 μm (inset A), or 50 μm (inset B). LP; lamina propria, C; caruncle.
(TIF)

**S3 Fig. Pathological manifestations in foetuses collected from RVFV infected ewes.** Healthy foetuses collected from ewes necropsied at one third (A) or at mid-gestation (B). (C) Foetuses carried by ewe 1764 that succumbed 4 days after inoculation with RVFV in experiment 1. (D) Live foetus collected from an ewe that was necropsied at 4 dpi in experiment 2. (E) Autolytic foetus collected from an ewe necropsied at 6 dpi in experiment 1. (F) Two aborted foetuses from experiment 2 (left) and one foetus (right) that was still inside the uterus at the moment of necropsy.
(TIF)

**S4 Fig. Schematic presentation of the ovine and human placenta.** A human placenta consists of a single discoid plaque whereas an ovine placenta consists of placentomes (A). A cross section of both placentas is depicted, showing the maternal tissues in shades of pink, and the foetal villi in orange. Blood and arteries are depicted in red, veins are depicted in blue (B). In the synepitheliochorial placenta (C, left panel), the foetal blood is separated from maternal blood by several maternal and foetal cell layers. In the haemophagous zone (C, middle panel) maternal blood is in direct contact with the foetal trophoblasts, which is similar to the human haemochorial placenta (C, right panel).
(TIF)

## Acknowledgments

We thank Pieter Roskam and Corry Dolstra (Wageningen Bioveterinary Research) for assisting with the necropsies and immunohistochemistry, respectively. We thank Dr. Schmaljohn (USAMRIID, Fort Detrick, MD) for providing the 4-D4 mAb. We also thank Dr. Guus Vermeulen for providing us with the human placentas.

## Author Contributions

**Conceptualization:** Judith Oymans, Lucien van Keulen, Jeroen Kortekaas.

**Funding acquisition:** Paul J. Wichgers Schreur, Jeroen Kortekaas.

**Investigation:** Judith Oymans, Lucien van Keulen, Jet Kant.

**Methodology:** Judith Oymans, Paul J. Wichgers Schreur, Lucien van Keulen, Jet Kant, Jeroen Kortekaas.

**Supervision:** Paul J. Wichgers Schreur, Lucien van Keulen, Jeroen Kortekaas.

**Writing – original draft:** Judith Oymans, Jeroen Kortekaas.

**Writing – review & editing:** Paul J. Wichgers Schreur, Lucien van Keulen.

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
