## [Decision Letter · Decision Letter 0]

27 Aug 2019

Dear Oymans:

Thank you very much for submitting your manuscript "Rift Valley fever virus targets the maternal-foetal interface in ovine and human placentas" (#PNTD-D-19-01103) for review by PLOS Neglected Tropical Diseases. Your manuscript was fully evaluated at the editorial level and by independent peer reviewers. The reviewers appreciated the attention to an important problem, but raised some substantial concerns about the manuscript as it currently stands. These issues must be addressed before we would be willing to consider a revised version of your study. We cannot, of course, promise publication at that time.

We therefore ask you to modify the manuscript according to the review recommendations before we can consider your manuscript for acceptance. Your revisions should address the specific points made by each reviewer. 

When you are ready to resubmit, please be prepared to upload the following:

(1) A letter containing a detailed list of your responses to the review comments and a description of the changes you have made in the manuscript.

(2) Two versions of the manuscript: one with either highlights or tracked changes denoting where the text has been changed (uploaded as a "Revised Article with Changes Highlighted" file); the other a clean version (uploaded as the article file).

(3) If available, a striking still image (a new image if one is available or an existing one from within your manuscript). If your manuscript is accepted for publication, this image may be featured on our website. Images should ideally be high resolution, eye-catching, single panel images; where one is available, please use 'add file' at the time of resubmission and select 'striking image' as the file type. 

Please provide a short caption, including credits, uploaded as a separate "Other" file. If your image is from someone other than yourself, please ensure that the artist has read and agreed to the terms and conditions of the Creative Commons Attribution License at http://journals.plos.org/plosntds/s/content-license (NOTE: we cannot publish copyrighted images). 

(4) If applicable, we encourage you to add a list of accession numbers/ID numbers for genes and proteins mentioned in the text (these should be listed as a paragraph at the end of the manuscript). You can supply accession numbers for any database, so long as the database is publicly accessible and stable. Examples include LocusLink and SwissProt.

(5) To enhance the reproducibility of your results, we recommend that you deposit your laboratory protocols in protocols.io, where a protocol can be assigned its own identifier (DOI) such that it can be cited independently in the future. For instructions see http://journals.plos.org/plosntds/s/submission-guidelines#loc-methods

While revising your submission, please upload your figure files to the Preflight Analysis and Conversion Engine (PACE) digital diagnostic tool, https://pacev2.apexcovantage.com/ PACE helps ensure that figures meet PLOS requirements. To use PACE, you must first register as a user. Then, login and navigate to the UPLOAD tab, where you will find detailed instructions on how to use the tool. If you encounter any issues or have any questions when using PACE, please email us at figures@plos.org.

We hope to receive your revised manuscript by Oct 26 2019 11:59PM. If you anticipate any delay in its return, we ask that you let us know the expected resubmission date by replying to this email.

To submit a revision, go to https://www.editorialmanager.com/pntd/ and log in as an Author. You will see a menu item call Submission Needing Revision. You will find your submission record there. 

Sincerely,

Abdallah M. Samy, PhD

Guest Editor

Paulo Pimenta

Deputy Editor

Reviewer's Responses to Questions

Key Review Criteria Required for Acceptance?

Methods

-Are the objectives of the study clearly articulated with a clear testable hypothesis stated?

-Is the study design appropriate to address the stated objectives?

-Is the population clearly described and appropriate for the hypothesis being tested?

-Is the sample size sufficient to ensure adequate power to address the hypothesis being tested?

-Were correct statistical analysis used to support conclusions?

-Are there concerns about ethical or regulatory requirements being met?

Reviewer #1: (No Response)

Reviewer #2: Yes

Results

-Does the analysis presented match the analysis plan?

-Are the results clearly and completely presented?

-Are the figures (Tables, Images) of sufficient quality for clarity?

Reviewer #1: Rift Valley fever virus (RVFV) is an arbovirus of the order Bunyavirales that causes severe disease in ruminants and humans. Surprisingly, pathology resulting in abortion remains poorly described. In the present work, pregnant ewes were inoculated intravenously with a highly virulent strain of RVFV and necropsied at different time points post inoculation. Tissues were collected and analyzed by real-time qPCR, virus isolation, and immunohistochemistry. Results reveal that maternal epithelial cells are the first placental cells to be infected, followed by fetal trophoblasts. In addition, RVFV can bypass the maternal epithelial cell layer by directly targeting fetal trophoblasts in the hemophagous zone, a region of the ovine placenta where maternal blood is in direct contact with fetal cells. Abortion was associated with widespread necrosis of placental tissues and severe hemorrhage. Experiments with human placental explants revealed efficient infection and replication in both cyto- and syncytiotrophoblasts. Results demonstrate that RVFV targets the fetal-maternal interface in both ovine and human placentas. The virus crossed the ovine placental barrier via two distinct routes, ultimately resulting in placental and fetal demise followed by abortion. Finding show that the same RVFV strain replicates efficiently in human trophoblasts and underscores the risk of RVFV infection during human pregnancy.

This is an excellently executed and well-illustrated study. Authors have nicely analyzed and compared ovine and human placental tissue with respect to RVFV infection.

It remains unclear how closely human placental explants mimic the in situ situation. In order to infect an ovine fetus a pathogen has to overcome several tissue layers compared to human placental tissue. This could indicate that human fetal infection should occur very frequently.

Authors report ….. “In addition, RVFV can bypass the maternal epithelial cell layer by directly targeting fetal trophoblasts in the hemophagous zone, a region of the ovine placenta where maternal blood is in direct contact with fetal cells.” Does this represents a back-door for any kind of placental infection in ruminants?

Conditions of RVFV PCR including used primers should be provided in more detail.

Table 1 should be accompanied by a more informative legend explaining the different colors, NT, abbreviations, mentioned internal case numbers, meaning of 1/3 and ½ gestation etc. Table should be readable without reading the remaining manuscript.

Numbering of figures starting at line 253 should occur in numeric/alphabetic order (e.g. please start with Figure 1a and not 1e).

It would be interesting to know whether alive fetuses of experiment 2 would develop long-term alterations in any other organ? 

How would the authors explain the difference with respect to placental/fetal susceptibility between experiment 1 and 2?

Line 272: it is stated: “…..observed an increase in mass of the fetal and maternal villi with concomitant decrease in mesenchyme (meaning mesenchymal tissue?)…….”. How was this statement substantiated (e.g. quantitative or semi-quantitative analysis?)? Furthermore, how would authors explain this tissue enlargement and decrease in mesenchymal tissue within such a short time period, especially in experiment 1? What about placental tissue of experiment 2 without macroscopic lesions? 

The question remains whether the extensive hemorrhage in placental tissue is solely due to a direct virus effect or represents a secondary event, at least in part, mediated by (maternal?) liver damage. 

Overall, I miss a comment on the pathology found in the ewes. It would be important to know whether animals suffered from severe liver damage, which could complicate or contribute to placental lesions. Furthermore, is there a preferential infection of placentomes during pregnancy or did animals suffer predominantly from liver necrosis and placental lesions represent a concurrent but less sever organ manifestation? 

It remains unclear whether abortion is due to placental necrosis, drop in systemic progesterone levels or systemic effects of RVFV infection in the ewe.

I miss a comment on the vasculature in the affected placentas (e.g. necrosis, vasculitis?). It is stated that…”blood vessels were found that stained positively for RVFV in the endothelium and/or smooth muscle cells in the blood vessel wall…”.in line 303/304. In addition, the prominent endothelial staining in figure 6 raises the question whether RVFV infection triggered morphologically recognizable or functional cell damage. Did these vessels also display necrosis (caspase 3, TUNEL-positivity?)?

Line 320: what is meant by …endothelial staining…(e.g. immunohistochemistry for endothelial cells or RVFV?)?

Minor points:

Figure 4: Please indicate (mark) more clearly the fetal and maternal part of the placentome.

Findings in figure 7 should be depicted at higher magnification for c and d, in addition. 

Blood-tingled number tags should be removed in the supplementary file figure 1 and 2.

Supplementary file figure 1 and 2: Please provide a more explanatory, descriptive legend.

Supplementary file figure 3b: Positive maternal vessels are not recognizable at the tip of the arrow. 

Mesenchyme is spelled differently in figure 3 and the text (line 273 and 374).

Reviewer #2: See my specific comments below regarding figures

Conclusions

-Are the conclusions supported by the data presented?

-Are the limitations of analysis clearly described?

-Do the authors discuss how these data can be helpful to advance our understanding of the topic under study?

-Is public health relevance addressed?

Reviewer #1: (No Response)

Reviewer #2: See my comments regarding speculation with regards to the conclusions

Editorial and Data Presentation Modifications?

Reviewer #1: (No Response)

Reviewer #2: Comments regarding figure and data clarity are below

Summary and General Comments

Reviewer #1: (No Response)

Reviewer #2: Livestock are highly susceptible to infection with Rift Valley fever virus (RVFV). Pregnant ewes suffer fetal abortions when infected in nature, and this massive fetal infection has been known since the time that RVF was first described. However, the mechanism by which RVFV infects the ovine placental has not been thoroughly documented in either a natural or experimental setting, and this manuscript addresses this important gap in the field. The authors perform experimental inoculation of pregnant sheep at 2 time points during gestation. The ewes and their fetuses are monitored for RVFV infection and pathology. The authors are to be commended for completion of a difficult study. Given the limitations of a large animal study with a high-containment agricultural pathogen, the data presented here contribute significantly to our understanding of livestock placental infection by RVFV. Comments are mostly minor for clarity of text and data presentation. This reviewer would caution the authors to avoid unnecessary speculation in the discussion. While the histology was limited to H&E and basic IHC, this reviewer feels the data largely support the conclusions. 

Specific comments:

1. Line 54 – Please clarify that the severe outcome of RVFV during LIVESTOCK pregnancy is well documented. The effect in human pregnancy is not as clear. 

2. Paragraph beginning at line 89 and ending at 105 describing the ovine placental structure should reference the appropriate citations. 

3. Line 143-44: Please clarify what “right upfront the national breeding season” means.

4. Line 171: Please clarify what 1% a/a means. 

5. Throughout the manuscript, the term “virus isolation” is used when it appears to refer to infectious virus quantification by TCID50. 

6. Results section beginning on line 235: Please provide information on how the inoculum dose and the gestation time points were chosen. 

7. Line 257: “dpi 7” should be “7 dpi”

8. Line 275-277: The authors refer to hemorrhage of the uterine wall and placentomes of ewe 1764– does this refer to macroscopic hemorrhage (as seen in Supp Fig 1) or microscopic hemorrhage visible by histology? 

9. Line 307 – remove the comma after Experiment 1.

10. Line 313-315 – Do the authors feel that the 4/8 dead fetuses without vRNA in the liver were not directly infected but rather died due to destruction of the associated placenta?

11. Lines 348-365 – Description of sheep pregnancy development needs the addition of references.

12. Line 356-365 – Since neither progesterone nor cytokine levels were measured in this study, the conclusions from this paragraph are speculative.

13. Lines 366-376 – Description of two routes that RVFV uses to cross the placenta is also largely speculative. The data in Fig 4 clearly show a lot of viral antigen within the placenta at 6 dpi, but without a true serial sacrifice experiment, it is difficult to definitively say what ‘route’ is used. 

Comments regarding figures:

1. Fig. 1 – please show rectal temperatures of the control group

2. Fig. 4 – What counterstain is used for the IHC staining? How certain are you of the trophoblast cell identification without cell-specific markers or a counterstain? It would be helpful to have a square or box in (a) and (d) indicating the zoomed-in areas shown in b, c, e, and f. It would also be helpful to have uninfected control placentomes stained for IHC visualize any background. In addition, labeling of the structures in Fig 4 (similar to the labeling in Fig 3c,d) would help to orient the reader since most readers are not intimately familiar with placental histology. There does not seem to be an arrow in 4b that is referred to in the legend.

3. Fig. 5 – There is no black asterisk that is referred to in the legend.

4. Fig. 7 – How many human placental donors were tested? How many tissue explants were tested per donor per time point? Is the data in this figure a compilation of replication from different donors or a single donor? What is the counterstain for the IHC? In c and d, are these tissue explants the cross section of a single villus? Please clarify what it is that the reader is looking at. In such a small piece of tissue, how confident are you of the cytotrophoblasts and syncytiotrophoblasts without a specific cell marker? Addition of arrows to indicate the different cell types would be helpful. Is there any background IHC staining in uninfected human placental explants? Control staining images would be helpful. 

5. Supplemental Fig 1 – are the negative control placenta and the RVFV-infected placenta from the same day of gestation?

6. Supplemental Fig 2 – same question for the control animals. Are they the same gestational age as the fetuses in c, d or e, f?

PLOS authors have the option to publish the peer review history of their article (what does this mean?). If published, this will include your full peer review and any attached files.

Do you want your identity to be public for this peer review? For information about this choice, including consent withdrawal, please see our Privacy Policy.

Reviewer #1: No

Reviewer #2: No

---

## [Decision Letter · Decision Letter 1]

1 Nov 2019

Dear Miss Oymans,

We are pleased to inform you that your manuscript, "Rift Valley fever virus targets the maternal-foetal interface in ovine and human placentas", has been editorially accepted for publication at PLOS Neglected Tropical Diseases.

Before your manuscript can be formally accepted and sent to production you will need to complete our formatting changes, which you will receive in a follow up email. Please note: your manuscript will not be scheduled for publication until you have made the required changes.

IMPORTANT NOTES

* Copyediting and Author Proofs: To ensure prompt publication, your manuscript will NOT be subject to detailed copyediting and you will NOT receive a typeset proof for review. The corresponding author will have one final opportunity to correct any errors when sent the requests mentioned above. Please review this version of your manuscript for any errors.

* If you or your institution will be preparing press materials for this manuscript, please inform our press team in advance at plosntds@plos.org. If you need to know your paper's publication date for media purposes, you must coordinate with our press team, and your manuscript will remain under a strict press embargo until the publication date and time. PLOS NTDs may choose to issue a press release for your article. If there is anything that the journal should know, please get in touch.

*Now that your manuscript has been provisionally accepted, please log into EM and update your profile. Go to http://www.editorialmanager.com/pntd, log in, and click on the "Update My Information" link at the top of the page. Please update your user information to ensure an efficient production and billing process.

*Note to LaTeX users only - Our staff will ask you to upload a TEX file in addition to the PDF before the paper can be sent to typesetting, so please carefully review our Latex Guidelines [http://www.plosntds.org/static/latexGuidelines.action] in the meantime.

Best regards,

Abdallah M. Samy, PhD

Guest Editor

Paulo Pimenta

Deputy Editor

---

## [Editor Report · Acceptance letter]

13 Jan 2020

Dear Miss Oymans,

We are delighted to inform you that your manuscript, "Rift Valley fever virus targets the maternal-foetal interface in ovine and human placentas," has been formally accepted for publication in PLOS Neglected Tropical Diseases.

Best regards,

Serap Aksoy

Editor-in-Chief

Shaden Kamhawi

Editor-in-Chief
